

# Diversity, taxonomic composition, and functional aspects of fungal communities in living, senesced, and fallen leaves at five sites across North America

Jana M. U'Ren[1,2] and A. Elizabeth Arnold[1,3]

[1] School of Plant Sciences, University of Arizona, Tucson, AZ, United States of America
[2] Department of Agricultural and Biosystems Engineering, University of Arizona, Tucson, AZ, United States of America
[3] Department of Ecology and Evolutionary Biology, University of Arizona, Tucson, AZ, United States of America

## ABSTRACT

**Background**. Fungal endophytes inhabit symptomless, living tissues of all major plant lineages to form one of earth's most prevalent groups of symbionts. Many reproduce from senesced and/or decomposing leaves and can produce extracellular leaf-degrading enzymes, blurring the line between symbiotrophy and saprotrophy. To better understand the endophyte–saprotroph continuum we compared fungal communities and functional traits of focal strains isolated from living leaves to those isolated from leaves after senescence and decomposition, with a focus on foliage of woody plants in five biogeographic provinces ranging from tundra to subtropical scrub forest.

**Methods**. We cultured fungi from the interior of surface-sterilized leaves that were living at the time of sampling (i.e., endophytes), leaves that were dead and were retained in plant canopies (dead leaf fungi, DLF), and fallen leaves (leaf litter fungi, LLF) from 3–4 species of woody plants in each of five sites in North America. Our sampling encompassed 18 plant species representing two families of Pinophyta and five families of Angiospermae. Diversity and composition of fungal communities within and among leaf life stages, hosts, and sites were compared using ITS-partial LSU rDNA data. We evaluated substrate use and enzyme activity by a subset of fungi isolated only from living tissues vs. fungi isolated only from non-living leaves.

**Results**. Across the diverse biomes and plant taxa surveyed here, culturable fungi from living leaves were isolated less frequently and were less diverse than those isolated from non-living leaves. Fungal communities in living leaves also differed detectably in composition from communities in dead leaves and leaf litter within focal sites and host taxa, regardless of differential weighting of rare and abundant fungi. All focal isolates grew on cellulose, lignin, and pectin as sole carbon sources, but none displayed ligninolytic or pectinolytic activity *in vitro*. Cellulolytic activity differed among fungal classes. Within Dothideomycetes, activity differed significantly between fungi from living vs. non-living leaves, but such differences were not observed in Sordariomycetes.

**Discussion**. Although some fungi with endophytic life stages clearly persist for periods of time in leaves after senescence and incorporation into leaf litter, our sampling across diverse biomes and host lineages detected consistent differences between fungal assemblages in living vs. non-living leaves, reflecting incursion by fungi from the leaf exterior after leaf death and as leaves begin to decompose. However, fungi found

Corresponding author
Jana M. U'Ren,
juren@email.arizona.edu

only in living leaves do not differ consistently in cellulolytic activity from those fungi detected thus far only in dead leaves. Future analyses should consider Basidiomycota in addition to the Ascomycota fungi evaluated here, and should explore more dimensions of functional traits and persistence to further define the endophytism-to-saprotrophy continuum.

## INTRODUCTION

Fungal endophytes inhabit symptomless, living photosynthetic tissues of all major plant lineages to form one of earth's most prevalent groups of symbionts (e.g., *Arnold et al., 2010*; *U'Ren et al., 2012*; *Zimmerman & Vitousek, 2012*; *Davey et al., 2013*; *Balínt et al., 2015*). Known from a wide range of biological provinces and agroecosystems, endophytes are a ubiquitous feature of plant biology (e.g., *Arnold & Lutzoni, 2007*; *Saunders, Glenn & Kohn, 2010*). Although classified together due to ecological similarities (i.e., colonization and transmission patterns, *in planta* biodiversity, and host interactions; see *Rodriguez et al., 2009*), endophytic fungi represent a diversity of evolutionary histories, life history strategies, and functional traits that are only beginning to be understood (reviewed by *Porras-Alfaro & Bayman, 2011*).

Endophytes that inhabit photosynthetic tissues of most plants are horizontally transmitted, form localized infections in aerial tissues, and represent highly diverse and often novel lineages (e.g., *Arnold et al., 2009*; *Gazis et al., 2012*; *Chen et al., 2015*; *U'Ren et al., 2016*) (Class 3 endophytes sensu *Rodriguez et al., 2009*; hereafter endophytes). Many reproduce from senesced and/or decomposing leaves (*Fröhlich & Hyde, 1999*; *Promputtha et al., 2007*; *Promputtha et al., 2010*; *U'Ren et al., 2010*; *Chaverri & Gazis, 2011*; *He et al., 2012*). Some also produce extracellular leaf-degrading enzymes (*Carroll & Petrini, 1983*; *Korkama-Rajala, Müller & Pennanen, 2008*; *Osono & Hirose, 2011*; *Promputtha et al., 2010*; *Sun, Guo & Hyde, 2011*). Thus many endophytes blur the line between symbiotrophy (during the endophytic phase) and saprotrophy (when they occur in association with dead tissue), creating a challenge for estimating trophic modes and the scale of fungal diversity based on species richness in particular functional groups.

The prevalence of saprotrophic life phases among endophytes and the dynamics of such fungi on the endophyte-to-saprotroph continuum are not yet clear. Current evidence suggests that different groups of fungi may persist for longer or shorter periods in senesced leaves, but the relevance of host lineages and abiotic factors is not yet understood in many cases. For example, *Osono (2006)* estimated that approximately two thirds of fungi with endophytic life stages can persist in and degrade leaf litter. *U'Ren et al. (2016)* found that 74% of Xylariaceae taxa were represented by isolates from asymptomatic plant tissues, as well as senesced/decomposing leaves, wood, bark, fruits, and/or flowers.

Some endophytes have ligninolytic and cellulolytic activity, cause mass loss from dead plant tissues *in vitro*, and persist as litter decomposers over multiple years (e.g., some *Lophodermium* spp. and xylariaceous fungi; see *Koide, Osono & Takeda, 2005*; *Osono, 2006*; *Korkama-Rajala, Müller & Pennanen, 2008*; *Osono & Hirose, 2009*; *Osono & Hirose, 2011*; see also *Lindahl et al., 2007*; *Yuan & Chen, 2014*). The presence of such fungi can increase respiration rates and lignocellulolytic activity in litter, altering the litter substrate and the activity of subsequent decomposers (see *Koide, Osono & Takeda, 2005*; *Šnajdr et al., 2011*; *He et al., 2012*; *Lin et al., 2015*). In contrast, other fungi with endophytic life phases may occupy litter only transiently, quickly sporulating from senesced and decomposing leaves to infect living tissues (e.g., *Rhabdocline parkeri* on *Pseudotsuga menziesii* and *Coccomyes nipponicum* on *Camellia japonica*; *Stone, 1987*; *Koide, Osono & Takeda, 2005*). The impact of such fungi on litter degradation is less well known. Such patterns can be documented via culture-based studies, but also by culture-free methods. For example, fungal genotypes that were dominant in the phyllosphere of living leaves of *Quercus petraea* disappeared by two to four months post abscission (*Voříšková & Baldrian, 2013*). Similarly, fungal communities in living leaves of European beech (*Fagus sylvatica*) in autumn differed from fungi inhabiting leaf litter and bark in the following spring (*Unterseher, Peršoh & Schnittler, 2013*; but see *Peršoh et al., 2013*).

Together, these studies speak to continuity between fungi that occur within living and non-living leaves (see also *Promputtha et al., 2007*; *Chaverri & Gazis, 2011*; *Sun, Guo & Hyde, 2011*; *Peršoh et al., 2013*; *Voříšková & Baldrian, 2013*; *Unterseher, Peršoh & Schnittler, 2013*). However, it is not clear whether such continuity is consistent in the distinctive plant and fungal communities present in different biomes, nor whether patterns of persistence vary among phylogenetically diverse plants. It is also unclear whether endophytes in a given plant species might occur in non-living leaves of co-occurring plant species: if undetected, such occurrences could alter our understanding of the endophyte–saprotroph continuum. Finally, little is known about functional traits of these fungi, raising the question: do strains that occur only living leaves differ in substrate use and enzyme activity from those that consistently occur in both living and non-living leaves?

The goal of this study was to explore the occurrence of endophytes in leaves after senescence and incorporation into the leaf litter, with a focus on woody plants in five biogeographic provinces ranging from tundra to subtropical scrub forest. Our surveys included >7,000 tissue pieces from living leaves, senesced leaves in plant canopies, and recently fallen leaf litter from 3–4 plant species in each site, yielding >2,000 fungal cultures. Here, we address (1) the degree to which fungal communities within leaves differ as a function of leaf type (living, senesced, or in leaf litter) at a given point in time; (2) whether such patterns are consistent among diverse biogeographic/bioclimatic zones and host lineages; and (3) how functional differences in carbon substrate utilization and enzyme activity reflect the occurrence of particular fungi in living vs. non-living leaves.

## MATERIALS AND METHODS

We collected living leaves, senesced leaves in plant canopies, and leaf litter from five sites representing distinct environmental, biological, and biogeographic regions of

North America: the Madrean Sky Island Archipelago of southeastern Arizona (AZC); the Appalachian Mountains of western North Carolina (NCH); sub-tropical scrub forest in Florida (FLA); Beringian tundra and boreal forest in the Seward Peninsula ecoregion of western Alaska (AKN); and inland, subalpine tundra in the Interior Highlands of east-central Alaska (AKE) (see *U'Ren et al., 2012* for site details). Within each site, we selected three to four species of woody plants that were representative of the community of the region (Table 1; *Hultén, 1965*; *Radford, Ahles & Bell, 1968*; *Barton, 1994*; http://www.archbold-station.org/html/datapub/species/lists/plantlist.html). Overall we examined 18 host species representing two families of Pinophyta and five families of Angiospermae (Table 1).

In each site we collected fresh tissues from one individual of each species in each of three replicate microsites located ca. 30 m apart along a 100 m transect, for a total of 9–12 host individuals per site. Within each microsite, focal plants occurred within close proximity to one another (<1–10 m apart), which allowed us to decouple spatial heterogeneity from host associations (*U'Ren et al., 2010*). The only exception was *Picea* in AKN (see Table 1; *U'Ren et al., 2012*).

From each individual we collected three small branches containing both healthy leaves and dead leaves attached to branches, as well as leaves in below-crown leaf litter in an intermediate state of decomposition (i.e., intact leaves with obvious changes in color and texture). Leaves of deciduous plants were selected to represent the same year of growth.

## Isolation of fungi

Plant material was transported in plastic bags to the laboratory and processed within 24 h (except AKN; 48 h) following *U'Ren et al. (2010)*. Each sample was washed thoroughly in running tap water for 30 sec. Although putatively saprotrophic fungi can be isolated from spores or hyphae on leaf surfaces (e.g., *Promputtha et al., 2002*; *Promputtha et al., 2007*), all leaves were sterilized to exclude fungi that were incidental on leaf surfaces and to maximize comparability with endophytic fungi (which are isolated from surface-sterilized leaves).

Leaves were cut into 2 mm$^2$ segments and surface-sterilized by agitating sequentially in 95% ethanol for 10 sec, 10% bleach (0.5 % NaOCl) for 2 min, and 70% ethanol for 2 min (*Arnold et al., 2007*). Segments were surface-dried under sterile conditions before being placed on 2% malt extract agar (MEA) in Petri dishes (16 tissue segments/dish) or 1.5 ml microcentrifuge tubes (1 segment/tube). Plates or tubes were sealed with Parafilm and incubated under ambient light/dark condition at room temperature (ca. 21.5 °C) for up to one year. Emergent fungi were isolated into pure culture, vouchered in sterile water, and deposited at the Robert L. Gilbertson Mycological Herbarium at the University of Arizona (ARIZ) (Table S1). Overall, 7,725 tissue segments were placed into culture (Table 2). All samples relevant to this work were handled in accordance with standard operating procedures for USDA permit P526P-1400151.

## DNA extraction, PCR, and sequencing

Total genomic DNA was extracted directly from each isolate following *Arnold & Lutzoni (2007)*. The nuclear ribosomal internal transcribed spacers and 5.8s gene

U'Ren and Arnold (2016), *PeerJ*, DOI 10.7717/peerj.2768

*Peer*J

**Table 1  Isolation frequency, richness and diversity of cultivable fungi from the interior of surface-sterilized leaves, including living leaves (i.e., endophytic fungi), fungi from dead leaves in canopies of woody plants (DLF), and fungi from leaf litter (LLF), from 18 plant species in five sites across North America.** Site abbreviations match those in *U'Ren et al. (2012)*: (AZC, Chiricahua Mountains, Arizona; NCH, Highlands Biological Station, North Carolina; FLA, Archbold Biological Station, Florida; AKE, Eagle Summit, Alaska; AKN, Nome, Alaska).

| Site | Host family | Host species[a] | Fungal type | Leaf segments[b] | Isolates recovered | Isolation frequency/ microsite ± SD | Isolates sequenced (%) | Basidio-mycota sequences | Ascomycota sequences (Putative species) | Ascomycota ACE (95% CI)[c] | Fisher's alpha |
|------|-------------|-----------------|-------------|------------------|--------------------|--------------------------------------|------------------------|--------------------------|------------------------------------------|----------------------------|----------------|
| AZC | Cupressaceae | *Juniperus deppeana* | endophyte | 144 | 22 | 0.15 ± 0.12 | 19 (86%) | 0 | 19 (10) | 27.2 (15.9, 16.3) | 8.5 |
| | | | DLF | 141 | 87 | 0.62 ± 0.18 | 82 (94%) | 0 | 82 (24) | 50.5 (36.2, 81.5) | 11.4 |
| | | | LLF | 144 | 43 | 0.30 ± 0.09 | 37 (86%) | 0 | 37 (18) | 22.8 (19.1, 38.0) | 13.8 |
| AZC | Pinaceae | *Pinus arizonica var. arizonica* | endophyte | 144 | 9 | 0.06 ± 0.02 | 9 (100%) | 0 | 9 (2) | 3.8 (2.2, 17.0) | 0.8 |
| | | | DLF | 144 | 36 | 0.25 ± 0.40 | 31 (86%) | 0 | 31 (3) | N/A | 0.8 |
| | | | LLF | 143 | 57 | 0.40 ± 0.35 | 55 (96%) | 0 | 55 (8) | 23.2 (13.5, 49.9) | 2.6 |
| AZC | Pinaceae | *Pseudotsuga menziesii* | endophyte | 144 | 21 | 0.15 ± 0.22 | 16 (76%) | 0 | 16 (3) | 4.0 (3.2, 7.1) | 1.1 |
| | | | DLF | 142 | 29 | 0.20 ± 0.21 | 19 (66%) | 0 | 19 (3) | NA | 1.0 |
| | | | LLF | 144 | 29 | 0.20 ± 0.03 | 25 (86%) | 0 | 25 (9) | 20.7 (12.7, 46.0) | 5.0 |
| AZC | Fagaceae | *Quercus rugosa* | endophyte | 130 | 0 | 0 | N/A | N/A | N/A | N/A | N/A |
| | | | DLF | 144 | 34 | 0.24 ± 0.39 | 32 (97%) | 0 | 32 (7) | N/A | 2.8 |
| | | | LLF | 143 | 29 | 0.20 ± 0.18 | 27 (90%) | 0 | 27 (7) | 9.5 (7.3, 25.3) | 3.1 |
| NCH | Pinaceae | *Pinus strobus* | endophyte | 114 | 14 | 0.14 ± 0.11 | 13 (93%) | 0 | 13 (6) | 30.6 (15.1, 72.5) | 4.3 |
| | | | DLF | 144 | 63 | 0.44 ± 0.31 | 61 (97%) | 1 | 60 (9) | 30.0 (10.6, 277.5) | 2.9 |
| | | | LLF | 144 | 62 | 0.43 ± 0.28 | 49 (79%) | 0 | 49 (16) | 25.9 (18.5, 55.2) | 8.3 |
| NCH | Pinaceae | *Tsuga canadensis* | endophyte | 144 | 54 | 0.38 ± 0.26 | 49 (91%) | 0 | 49 (13) | 19 (16.5, 23.4) | 5.8 |
| | | | DLF | 144 | 53 | 0.37 ± 0.39 | 42 (79%) | 0 | 42 (24) | 162.2 (108.9, 248.8) | 23.3 |
| | | | LLF | 144 | 123 | 0.85 ± 0.08 | 95 (77%) | 2 | 93 (26) | 48.0 (35.8, 75.4) | 12.0 |

U'Ren and Arnold (2016), *PeerJ*, DOI 10.7717/peerj.2768

**Table 1** (*continued*)

| Site | Host family | Host species[a] | Fungal type | Leaf segments[b] | Isolates recovered | Isolation frequency/microsite ± SD | Isolates sequenced (%) | Basidio-mycota sequences | Ascomycota sequences (Putative species) | Ascomycota ACE (95% CI)[c] | Fisher's alpha |
|---|---|---|---|---|---|---|---|---|---|---|---|
| NCH | Ericaceae | *Kalmia latifolia* L. | endophyte | 144 | 29 | 0.20 ± 0.31 | 29 (100%) | 0 | 29 (6) | 9.8 (7.6, 15.0) | 2.3 |
| | | | DLF | 144 | 71 | 0.49 ± 0.34 | 65 (92%) | 0 | 65 (14) | 43.4 (26.6, 82.4) | 5.5 |
| | | | LLF | 144 | 114 | 0.79 ± 0.15 | 91 (80%) | 0 | 91 (18) | 66.1 (41.3, 117.5) | 6.7 |
| NCH | Fagaceae | *Quercus montana* | endophyte | 144 | 60 | 0.42 ± 0.33 | 57 (95%) | 0 | 57 (6) | 25.2 (11.5, 73.1) | 1.7 |
| | | | DLF | 144 | 58 | 0.40 ± 0.42 | 47 (81%) | 0 | 47 (12) | 170.2 (102.8, 287.5) | 5.2 |
| | | | LLF | 144 | 46 | 0.32 ± 0.17 | 27 (59%) | 0 | 27 (13) | 38.9 (23.6, 76.4) | 9.9 |
| FLA | Pinaceae | *Pinus elliottii* | endophyte | 144 | 138 | 0.96 ± 0.04 | 130 (94%) | 57 | 73 (32) | 60.6 (42.5, 109.9) | 21.7 |
| | | | DLF | 144 | 102 | 0.71 ± 0.29 | 84 (82%) | 3 | 81 (32) | 126.6 (82.0, 210.9) | 19.5 |
| | | | LLF | 144 | 100 | 0.69 ± 0.23 | 85 (85%) | 0 | 85 (28) | 53.0 (36.6, 100.8) | 14.6 |
| FLA | Pinaceae | *Pinus clausa* | endophyte | 144 | 52 | 0.36 ± 0.07 | 47 (90%) | 29 | 18 (11) | 111.1 (59.3, 218.8) | 12.0 |
| | | | DLF | 144 | 88 | 0.61 ± 0.33 | 75 (85%) | 2 | 73 (30) | 88.9 (61.5, 140.2) | 19.0 |
| | | | LLF | 144 | 104 | 0.72 ± 0.10 | 88 (85%) | 3 | 85 (23) | 38.0 (27.6, 71.9) | 10.4 |
| FLA | Arecaceae | *Serenoa repens* | endophyte | 144 | 65 | 0.45 ± 0.40 | 58 (89%) | 17 | 41 (10) | 17.4 (12.1, 35.9) | 4.2 |
| | | | DLF | 144 | 81 | 0.56 ± 0.21 | 56 (69%) | 1 | 55 (33) | 154.3 (105.4, 236.4) | 34.8 |
| | | | LLF | 144 | 73 | 0.51 ± 0.17 | 58 (79%) | 2 | 56 (39) | 70.5 (62.3, 81.5) | 57.0 |
| FLA | Fagaceae | *Quercus inopina* | endophyte | 144 | 80 | 0.56 ± 0.25 | 51 (64%) | 1 | 50 (19) | 89.0 (55.7, 152.6) | 11.2 |
| | | | DLF | 144 | 95 | 0.66 ± 0.50 | 67 (71%) | 0 | 67 (28) | 119.4 (81.4, 184.5) | 18.1 |
| | | | LLF | 144 | 100 | 0.69 ± 0.35 | 72 (72%) | 0 | 72 (41) | 153.6 (107.1, 232.9) | 39.5 |

U'Ren and Arnold (2016), *PeerJ*, DOI 10.7717/peerj.2768

**Table 1** (*continued*)

| Site | Host family | Host species[a] | Fungal type | Leaf segments[b] | Isolates recovered | Isolation frequency/ microsite ± SD | Isolates sequenced (%) | Basidio-mycota sequences | Ascomycota sequences (Putative species) | Ascomycota ACE (95% CI)[c] | Fisher's alpha |
|---|---|---|---|---|---|---|---|---|---|---|---|
| AKE | Pinaceae | *Picea glauca* | endophyte | 144 | 8 | 0.06 ± 0.03 | 7 (88%) | 1 | 6 (6) | N/A | N/A |
| | | | DLF | 144 | 43 | 0.30 ± 0.16 | 36 (84%) | 0 | 36 (17) | 48.8 (31.3, 87.7) | 12.6 |
| | | | LLF | 144 | 77 | 0.53 ± 0.32 | 64 (83%) | 1 | 63 (29) | 51.1 (36.9, 91.1) | 20.8 |
| AKE | Betulaceae | *Betula nana* | endophyte | 144 | 2 | 0.01 ± 0.01 | 2 (100%) | 0 | 2 (2) | N/A | N/A |
| | | | DLF | 144 | 10 | 0.07 ± 0.05 | 10 (100%) | 0 | 10 (6) | 65.7 (28.8, 162.5) | 6.3 |
| | | | LLF | 144 | 37 | 0.26 ± 0.25 | 36 (97%) | 0 | 36 (18) | 48.0 (30.9, 88.0) | 14.3 |
| AKE | Salicaceae | *Salix pulchra* | endophyte | 144 | 3 | 0.02 ± 0.02 | 2 (67%) | 0 | 2 (2) | N/A | N/A |
| | | | DLF | 144 | 15 | 0.10 ± 0.04 | 12 (80%) | 0 | 12 (9) | 26.8 (12.3, 104.3) | 16.4 |
| | | | LLF | 144 | 19 | 0.13 ± 0.10 | 13 (68%) | 0 | 13 (9) | 14.6 (12.9, 17.1) | 12.9 |
| AKN | Pinaceae | *Picea glauca*[a] | endophyte | 144 | 68 | 0.47 ± 0.37 | 65 (96%) | 0 | 65 (8) | 9.2 (8.1, 18.8) | 2.4 |
| | | | DLF | 144 | 75 | 0.52 ± 0.15 | 60 (80%) | 0 | 60 (20) | 42.0 (29.6, 70.1) | 10.5 |
| | | | LLF | 144 | 35 | 0.24 ± 0.23 | 32 (91%) | 5 | 27 (11) | 17.9 (12.4, 45.0) | 6.9 |
| AKN | Salicaceae | *Salix pulchra* | endophyte | 144 | 0 | 0 | N/A | N/A | N/A | N/A | N/A |
| | | | DLF | 144 | 0 | 0 | NA | N/A | N/A | N/A | N/A |
| | | | LLF | 144 | 1 | 0.01 ± 0.01 | | N/A | N/A | N/A | N/A |
| AKN | Betulaceae | *Betula nana* | endophyte | 144 | 0 | 0 | N/A | N/A | N/A | N/A | N/A |
| | | | DLF | 144 | 2 | 0.01 ± 0.02 | 1 (50%) | 0 | 1 (1) | N/A | N/A |
| | | | LLF | 144 | 2 | 0.01 ± 0.01 | 1 (50%) | 0 | 1 (1) | N/A | N/A |
| Total | | | | 7,725 | 2618 | 0.34 ± 0.26 | 2189 (84%) | 125 | 2064 (306) | 437.8 (395.7, 499.9) | 99.3 |

**Notes.**

[a]Hosts were collected approximately 60 km east of Nome, in Council, AK (the nearest site that contains trees).

[b]We initially sampled 144 leaf segments from each host species in each site (i.e., 48 per microsite for three microsites), but a small number were lost to contamination, overgrowth, and desiccation.

[c]The diversity of Ascomycota sequences was estimated with ACE (abundance-based coverage estimator).

Peerj

**Table 2** Summary of isolation frequency, richness and diversity of endophytic fungi, fungi from dead leaves in tree canopies (DLF), and fungi from leaf litter (LLF) in five North American sites.

| | Site | Host species (yielding cultures) | Isolates | Isolation frequency ± SD[*] | Isolates sequenced (%) | Basidiomycota sequences | Ascomycota sequences | Ascomycota putative species (95% CI) | Richness estimates (Bootstrap, ACE) | Fisher's alpha (FA) | Mean FA/host species ± SD[**] |
|---|---|---|---|---|---|---|---|---|---|---|---|
| Endophyte | AZC | 4 (3) | 52 | $0.09 \pm 0.07^{ab}$ | 44 (85%) | 0 | 44 | 15 (10.9, 19.1) | 17.2, 17.9 | 8 | $3.5 \pm 4.4$ |
| | NCH | 4 (4) | 157 | $0.28 \pm 0.14^{ab}$ | 148 (94%) | 0 | 148 | 27 (25.1, 28.9) | 29.6, 29.3 | 9.7 | $3.5 \pm 1.9$ |
| | FLA | 4 (4) | 335 | $0.58 \pm 0.26^{a}$ | 286 (85%) | 104 | 182 | 60 (51.7, 68.3) | 71.7, 95.4 | 31.2 | $12.3 \pm 7.2$ |
| | AKE | 3 (3) | 13 | $0.03 \pm 0.02^{b}$ | 11 (85%) | 1 | 10 | 10 (4.5, 15.6) | 13.5, 55.0 | N/A | N/A |
| | AKN | 3 (1) | 68 | $0.16 \pm 0.27^{ab}$ | 65 (96%) | 0 | 65 | 8 (5.5, 10.5) | 8.9, 10.0 | 2.4 | 2.4 |
| | Total | 18 (15) | 625 | $0.24 \pm 0.26^{B}$ | 554 (89%) | 105 | 449 | 108 (99.6, 116.1) | 122.6, 128.5 | 45.1 | $6.3 \pm 6.2^{B}$ |
| DLF | AZC | 4 (4) | 186 | $0.33 \pm 0.19^{ab}$ | 164 (88%) | 0 | 164 | 33 (28.3, 37.7) | 37.8, 42.9 | 12.4 | $4.0 \pm 5.0^{b}$ |
| | NCH | 4 (4) | 245 | $0.43 \pm 0.05^{ab}$ | 215 (88%) | 1 | 214 | 50 (42.1, 57.9) | 57.6, 67.1 | 20.5 | $9.22 \pm 9.4^{ab}$ |
| | FLA | 4 (4) | 366 | $0.64 \pm 0.06^{a}$ | 282 (77%) | 6 | 276 | 81 (69.1, 92.9) | 95.9, 122.3 | 38.6 | $22.9 \pm 8.0^{a}$ |
| | AKE | 3 (2) | 68 | $0.16 \pm 0.12^{b}$ | 58 (85%) | 0 | 58 | 25 (18.7, 31.3) | 30.1, 38.6 | 16.7 | $11.8 \pm 5.1^{ab}$ |
| | AKN | 3 (3) | 77 | $0.18 \pm 0.30^{b}$ | 61 (79%) | 0 | 61 | 21 (16.3, 25.7) | 25.1, 33.0 | 11.3 | 10.5 |
| | Total | 18 (17) | 942 | $0.36 \pm 0.23^{A}$ | 780 (83%) | 7 | 773 | 182 (168.7, 195.3) | 211, 240.9 | 75.1 | $11.9 \pm 9.5^{AB}$ |
| LLF | AZC | 4 (4) | 158 | $0.28 \pm 0.09^{ab}$ | 144 (91%) | 0 | 144 | 35 (29.7, 40.2) | 40.8, 46.4 | 14.7 | $6.1 \pm 5.2^{b}$ |
| | NCH | 4 (4) | 345 | $0.60 \pm 0.26^{a}$ | 262 (76%) | 2 | 260 | 58 (49.8, 66.3) | 65, 67.0 | 23.2 | $9.2 \pm 2.3^{ab}$ |
| | FLA | 4 (4) | 377 | $0.66 \pm 0.10^{a}$ | 303 (80%) | 5 | 298 | 93 (84.1, 101.9) | 109.3, 125.0 | 46.4 | $30.4 \pm 21.9^{a}$ |
| | AKE | 3 (3) | 133 | $0.31 \pm 0.20^{ab}$ | 113 (85%) | 1 | 112 | 41 (33.3, 48.7) | 49.6, 70.7 | 23.3 | $16 \pm 4.2^{ab}$ |
| | AKN | 3 (3) | 38 | $0.09 \pm 0.13^{b}$ | 33 (90%) | 5 | 28 | 12 (9.7, 14.3) | 14.1, 15.0 | 8 | 6.9 |
| | Total | 18 (18) | 1051 | $0.41 \pm 0.26^{A}$ | 855 (81%) | 13 | 842 | 195 (179.1, 210.9) | 226.9, 266.3 | 79.6 | $14.9 \pm 14.2^{A}$ |

**Notes.**

[*]Different letters represent significant differences ($P < 0.05$) in isolation frequency among sites for each leaf type (lowercase; endophytic: ANOVA $F_{4,10} = 7.31$, $P = 0.0051$; DLF: ANOVA $F_{4,12} = 3.30$, $P = 0.0482$; LLF: ANOVA $F_{4,13} = 7.41$, $P = 0.0025$) based on post-hoc Tukey's HSD comparisons and among leaf types (uppercase: linear contrasts following ANOVA on residuals after test for site effects; $F_{1,47} = 4.11$, $P = 0.0484$).

[**]Different letters represent significant differences ($P < 0.05$) in diversity per-host-species among leaf types (uppercase; ANOVA on residuals following test for site effects, $F_{4,39} = 11.56$, $P < 0.0001$) and among sites for DLF and LLF (lowercase; DLF: ANOVA on log FA, $F_{3,11} = 5.35$ $P = 0.0162$; LLF: ANOVA on log FA, $F_{3,11} = 5.27$ $P = 0.0170$), but endophytic fungi only approached significance (ANOVA on log FA, $F_{2,8} = 3.87$ $P = 0.0667$).

(ITS rDNA; ca. 600 bp) and an adjacent portion of the nuclear ribosomal large subunit (LSU rDNA; ca. 500 bp) were amplified by PCR as a single fragment using primers ITS1F or ITS5 and LR3 (*White et al., 1990*; *Gardes & Bruns, 1993*; *Vilgalys & Hester, 1990*; ITS rDNA-partial LSU rDNA) following *U'Ren et al. (2010)*. ITS rDNA was amplified using primers ITS1 and ITS4 (*White et al., 1990*) for 19 isolates that failed to amplify using the primers listed above. PCR conditions are described in *U'Ren et al. (2010)*.

PCR products were evaluated by staining with SYBR Green I (Molecular Probes, Invitrogen; Carlsbad, CA, USA) after electrophoresis on a 1% agarose gel. All positive amplicons yielded single bands. PCR products were cleaned, quantified, normalized, and sequenced directly with the Applied Biosystems BigDye® Terminator v3.1 cycle sequencing kit and the original PCR primers at the University of Arizona Genetics Core. Bidirectional sequencing was performed on an Applied Biosystems 3730*xl* DNA Analyzer (Foster City, CA, USA).

The software applications *phred* and *phrap* (*Ewing & Green, 1998*; *Ewing et al., 1998*) were used to call bases and assemble contigs with automation provided by the ChromaSeq package in Mesquite v. 1.06 (http://mesquiteproject.org). Base calls were verified by inspection of chromatograms in Sequencher v. 4.5 (Gene Codes, Ann Arbor, MI, USA). Overall, 1,037 new sequences were analyzed in conjunction with 1,030 sequences previously published as part of a study of endophytic and endolichenic fungal communities (*U'Ren et al., 2010*; *U'Ren et al., 2012*; *U'Ren et al., 2016*). Each sequence was queried against the non-redundant UNITE+INSDC database (v. 7.97.31.01, released 2016-01-31) using the Ribosomal Database Project (RDP) naïve Bayesian classifier (*Wang et al., 2007*) with a confidence threshold cutoff of 80% to estimate taxonomic placement. Basidiomycota comprised <6% of the overall data set and were excluded from all analyses due to their rarity, potentially reflecting use of a culture medium demonstrated to recover a high diversity of ascomycetous endophytes (*Fröhlich & Hyde, 1999*; *Arnold, 2002*). All sequences have been deposited in GenBank (Table S2).

## Species richness and diversity

Operational taxonomic units (OTU) were defined using ITS-partial LSU rDNA sequences. To designate OTU a distance matrix based on pairwise Needleman-Wunsch alignments was generated with the needledist module in ESPRIT (*Sun et al., 2009*) under the default parameters. Gaps of any length were treated as a single evolutionary event and terminal gaps were not penalized (*Sun et al., 2009*). We used mothur (*Schloss et al., 2009*) to cluster sequences into OTU using the complete-linkage method (i.e., furthest neighbor) and to calculate richness and OTU overlap among communities. Sequences were assembled into OTU based on sequence similarities ranging from 95% to 100% following *U'Ren et al. (2012)*. Results are presented at the genotype level (i.e., 100% sequence similarity) and at the putative species level (i.e., 95% sequence similarity; see *U'Ren et al., 2009*). Previous assessment of four endophyte-rich genera in the Sordariomycetes and Dothideomycetes demonstrated that ca. 5% ITS rDNA divergence (i.e., 95% sequence similarity) conservatively estimates sister species boundaries when compared against published phylogenies (*U'Ren et al., 2009*; see also *Liggenstoffer et al., 2010*). OTU richness

increased when more stringent sequence similarity levels were used (e.g., 348 OTU were estimated at 97% sequence similarity vs. 306 OTU at 95% sequence similarity), generally reflecting an increase in singletons that could not be used for comparative analyses. Even so, estimates of overlap among different leaf types were congruent based on 95–100% sequence similarity groups.

OTU accumulation curves (Fig. 1; Fig. S1), rarefaction analyses, estimates of total richness (bootstrap and abundance-based coverage estimator (ACE), recommended by *Hortal, Borges & Gaspar, 2006*), and diversity were inferred in EstimateS v. 8.0 (http://viceroy.eeb.uconn.edu/EstimateS) using 50 randomizations of sample order without replacement. Fisher's alpha, a parameter of the log series model that is theoretically independent of sample size (*Fisher, Corbet & Williams, 1943*; *Taylor, 1978*; *Magurran, 2004*), was chosen to measure fungal diversity. Fisher's alpha is robust for comparing diversity among communities that are sampled unequally (given variation in isolation frequency) and are characterized by a log-series distribution of OTU abundance (*Magurran, 2004*).

We analyzed isolation frequency and diversity as a function of site, host family, and leaf type (i.e., living, senesced, or leaf litter) using multiple regression with these explanatory factors as main effects. Isolation frequency and diversity were logit and log transformed, respectively, to achieve normality. The fit of each model was assessed using a lack-of-fit F test. Pairwise comparisons between fungal communities from different leaf types were computed using least-squares means contrasts. Analyses were done in JMP v. 9.0.0 (SAS Institute, Cary, NC).

## Similarity of fungal assemblages

We used several approaches to examine the relationship of site, host, and leaf type to the composition of fungal communities. First, we used analysis of similarity (ANOSIM) coupled with visualization by non-metric multidimensional scaling (NMDS) to assess whether fungal communities differed significantly among sites (Fig. S2). Because fungal communities did differ among sites, NMDS ordinations also were computed within each site to examine the effect of host species and leaf type (Fig. S2). ANOSIM and NMDS were conducted in PAST v. 1.88 (*Hammer, Harper & Ryan, 2001*) following *U'Ren et al. (2012)*.

Second, these analyses were complemented by hierarchical clustering to clarify the relationships among fungal communities in different sites and leaf types. The dendrogram was generated using the unweighted pair-group average (UPGMA) algorithm. Hierarchical clustering was conducted in PAST v. 1.88 (*Hammer, Harper & Ryan, 2001*) using 10,000 bootstrap replicates.

Third, permutational multivariate analysis of variance (PERMANOVA) was used to examine variation in community composition as a function of (1) leaf type and host species, while constraining permutations with each site; and (2) leaf type, while constraining permutations with each site and each host species. PERMANOVA was implemented with the adonis and permute functions in the R (R Core Team) package vegan (*Oksanen et al., 2015*). ANOSIM, NMDS, hierarchical clustering, and PERMANOVA were conducted using all non-singleton OTU (i.e., those occurring more than once in the entire

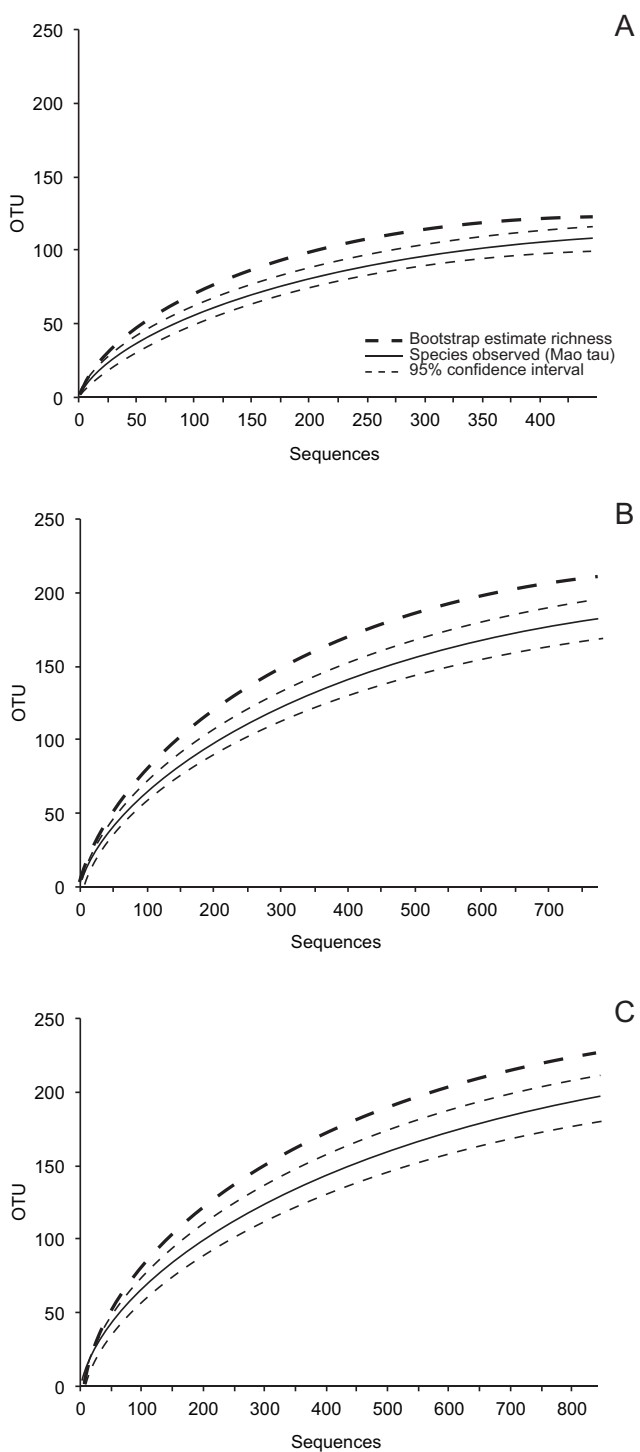

**Figure 1** **Species accumulation curves (Mao Tau), 95% confidence intervals, and bootstrap estimates of richness based on ITS-partial LSU rDNA.** (A) 449 isolates of endophytic fungi; (B) 773 isolates of fungi from senesced leaves in plant canopies (dead leaf fungi, DLF); and (C) 842 isolates of fungi from leaf litter (leaf litter fungi, LLF) in five sites.

dataset) to minimize the potential under-estimation of similarity due to undersampling (*Manter & Bakker, 2015*). The Morisita-Horn metric of similarity was used to minimize sensitivity to under-sampling, as it reflects the distribution of abundant taxa (*Beck, Holloway & Schwanghart, 2013*). Host individuals/leaf types from which <4 isolates were sequenced were excluded from all analyses (see Table 1).

Finally, because sample sizes generally were low for each host species-leaf type combination, and because distributions of rare taxa can be informative, we examined community structure for all OTU grouped by site and leaf type using OTUshuff (*Manter & Bakker, 2015*). To estimate the influence of rare and abundant taxa on community similarity, OTUshuff uses a variably weighted version of the Odum index (wOdum). At $\alpha = 1$, $D_{wOdum}$ is equal to $D_{Odum}$; when $\alpha < 1$, the influence of abundant taxa will be down-weighted; and when $\alpha > 1$, the influence of abundant abundance taxa will be up-weighted. We varied the alpha weighting factor from 0 to 4 to test the null hypothesis that samples from different leaf types within a site represent the same community (Table S1). Analyses were conducted using a Monte Carlo simulation with 1,000 iterations. Low isolation frequency of endophytes prevented comparisons of endophytes and DLF/LLF in one site (AKE).

## Functional characterization of fungal isolates

We compared *in vitro* growth and cellulolytic, pectinolytic, and ligninolytic activity for representative isolates of eight OTU found only in living leaves and nine OTU found only in non-living leaves. Each OTU was represented by a single isolate. Leaf type designation for each OTU (i.e., living leaves vs. non-living leaves) was based on host and substrate information from a larger collection of endophytic and endolichenic fungi (see *U'Ren et al., 2012*), as well as BLASTn comparisons to the NCBI nr database. When possible, OTU representing each leaf status (living, non-living) were chosen from the same class (i.e., Sordariomycetes and Dothideomycetes). However, paired comparisons were not possible for Pezizomycetes or Leotiomycetes because no OTU in those classes were found only as endophytes.

Pure cultures were started on 2% MEA plates and were allowed to grow for up to 7 d. At that point 5 mm agar plugs containing actively growing mycelium were placed on water agar plates (WA; pH 5.6, 1.5% agar, containing 50 ml 20X nitrate salts, 1 ml 1000X trace elements, and 1 ml vitamin solution per liter; see *Tucker & Orbach, 2007*) amended with (1) 5 g carboxymethylcellulose (cellulose substitute; Sigma-Aldrich, St. Louis, Missouri, USA); (2) 10 g citrus pectin (grade 1 from citrus fruits, Sigma-Aldrich); or (3) 0.5 g indulin (lignin substitute; Mead-Westvaco, Richmond, Virginia, USA) as the sole carbon source. Three replicates were prepared for each isolate per medium. A culture with verified enzymatic activity was used as the positive control.

Cultures were grown for 5–28 days in a temperature controlled room at 26 °C under constant light conditions. Once cultures were ca. 3–4 cm in diameter, plates were stained with aqueous solutions of 0.2 % (wt/vol) Congo red, equal parts of 1.0 % (wt/vol) $FeCl_3$ and $K_3[Fe(CN)]_6$, or 0.05% (wt/vol) ruthenium red to detect cellulolytic, pectinolytic, and ligninolytic activity, respectively (modified from *Gazis et al., 2012*; *Cotty et al., 1990*).

Following staining, plates were rinsed with either 1M sodium chloride (for cellulose plates) or deionized water (pectin and lignin plates) and the colony diameter and clearing at the periphery of mycelial growth (i.e., extra-hyphal clearing) of each isolate was measured.

Average growth rates (colony diameter/days of growth) were compared on each carbon source as a function of fungal taxonomy at the class level and leaf type using non-parametric tests. The proportion of isolates that demonstrated diagnostic clearing was compared as a function of leaf type using two-tailed Fisher's exact tests. Where present, the amount of clearing at the periphery of mycelial growth was scaled by growth rate and compared using Wilcoxon rank sum tests.

Mantel tests were used to examine the relationship of fungal genetic distance to patterns of cellulolytic activity and growth rates on different carbon sources. ITS-partial LSU rDNA sequences for Sordariomycetes and Dothideomycetes isolates were aligned using MAFFT v6.821b (*Katoh & Toh, 2008*) with the L-INS-I setting for high accuracy. Ambiguously aligned nucleotides were masked using the Gblocks server v.0.91b (*Castresana, 2000*) with parameters to allow for less stringent selection (i.e., to allow for smaller final blocks; to allow gap positions within the final blocks; and to allow less strict flanking positions) resulting in retention of 83% of the original 1,223 positions. A distance matrix was calculated in PAUP* v4.0 (*Swofford, 2003*) using uncorrected distances (p). Pairwise distance matrices for (1) cellulolytic activity; (2) average growth rate on cellulose; (3) average growth rate on pectin; and (4) average growth rate on lignin were calculated in JMP v. 9.0.0. Mantel tests were performed in R with the ade4 package (*Dray & Dufour, 2007*) using 9,999 permutations.

## RESULTS

We obtained 2,618 fungal isolates from the interior of 7,725 leaf segments representing living leaves (i.e., endophytes), dead leaves in plant canopies (dead leaf fungi, DLF), and leaf litter (leaf litter fungi, LLF; Table 1) of 18 host species. ITS rDNA partial LSU rDNA data obtained from 2,064 isolates of Ascomycota comprised 555 unique genotypes (100% sequence similarity) and 306 putative species (i.e., OTU at 95% sequence similarity) (Table 1).

Our sampling effort did not capture the estimated richness of cultivable leaf-associated fungi for the entire study (Fig. 1; Table 2). However, sampling in eight of 15 site-leaf type combinations encompassed the local richness of fungi based on bootstrap estimates (Table 2), and average differences between bootstrap and ACE estimates and the upper 95% confidence intervals around observed richness were only 3.0 and 13.7 OTU, respectively (Table 2; see also Fig. S1). Therefore, we used these data to compare communities of culturable fungi as described below.

### Isolation frequency and diversity

Isolation frequency and diversity differed as a function of site, host family, and leaf type (Table 3). Across all sites, fungi were isolated less frequently from living leaves than from dead leaves in plant canopies or leaf litter (Table 3). Endophytes were isolated from 24% ± 26% (mean ± SD) of leaf segments per host species (range: 0–96%; Table 2). DLF and LLF were isolated from 36% ± 23% (range, 0–71%) and 41% ± 26% (range, 1–85%) of

**Table 3** Statistical tests of isolation frequency and diversity in multiple regression models.

| | Isolation frequency ($R^2 = 0.73$, $P < 0.0001$) | | | Diversity ($R^2 = 0.80$, $P < 0.0001$) | | |
|---|---|---|---|---|---|---|
| | DF | F | P | DF | F | P |
| Explanatory variables[a] | | | | | | |
|   Site | 4 | 7.07 | 0.0002 | 4 | 20.18 | <0.0001 |
|   Host family | 6 | 5.15 | 0.0006 | 6 | 4.8 | 0.0014 |
|   Leaf type | 2 | 4.52 | 0.0174 | 2 | 10.76 | 0.0003 |
| Leaf type[b] | | | | | | |
|   Endo vs. DLF | 1 | 5.46 | 0.0250 | 1 | 9.53 | 0.0042 |
|   Endo vs. LLF | 1 | 8.19 | 0.0069 | 1 | 21.34 | <0.0001 |
|   DLF vs. LLF | 1 | 0.29 | 0.5943 | 1 | 2.9 | 0.0986 |

**Notes.**
[a] Explanatory variables: Effect test of site, host family and leaf type.
[b] Leaf type: Least-squares means contrasts, pairwise comparison among endophytes, DLF, and LLF.

leaf segments per host species, respectively (Table 1). A greater number of hosts failed to yield cultures from living leaves (i.e., *Salix pulchra* and *Betula nana* at AKN and *Quercus rugosa* at AZC; Table 2) compared to a single host for non-living leaves (*Betula nana* at AKN: only LLF were obtained; Table 1).

Endophytes also were less diverse than DLF and LLF (Table 3). Cumulative diversity of fungi from non-living leaves over the entire study was 1.7 to 1.8 times that of fungi from living leaves (Table 2).

## Fungal community analyses

Fungal communities differed among sites (Fig. 2; Fig. S2A). Within sites, communities differed among host species and leaf type (Fig. 2; Table 4; Fig. S2B–S2D).

Within each site, we rejected the null hypothesis that endophytes and DLF/LLF were from the same community at all levels of alpha ($P < 0.05$; Table S1). Preferentially weighting rare species decreased similarity between endophytes and DLF, and between DLF and LLF, relative to analyses in which abundant species were weighted preferentially. In contrast, variably weighting rare or abundant OTU had no discernible effect on similarity estimates of endophytes and LLF, which were consistently low (Fig. 3; Table S1).

Overall, 29% of non-singleton OTU (Fig. S3) and 52% of common OTU (those represented by >5 sequenced isolates) were found in both living and non-living leaves (Fig. 4). After removing infrequent taxa, OTU from only one leaf status were rare: only 3%, 5%, or 8% of OTU were unique to living leaves, dead leaves in plant canopies, or leaf litter, respectively (Fig. 4; see also Fig. S4). However, OTU that were highly abundant in leaf litter were seldom found as endophytes (Fig. 4), and the abundance of OTU from leaf litter was not correlated with abundance of the same OTU in living leaves (Table S3).

## Substrate utilization

We compared *in vitro* growth for representative isolates of eight OTU found only in living leaves and nine OTU found only in non-living leaves. All isolates, regardless of leaf type of origin, grew on cellulose, lignin, and pectin as sole carbon sources.

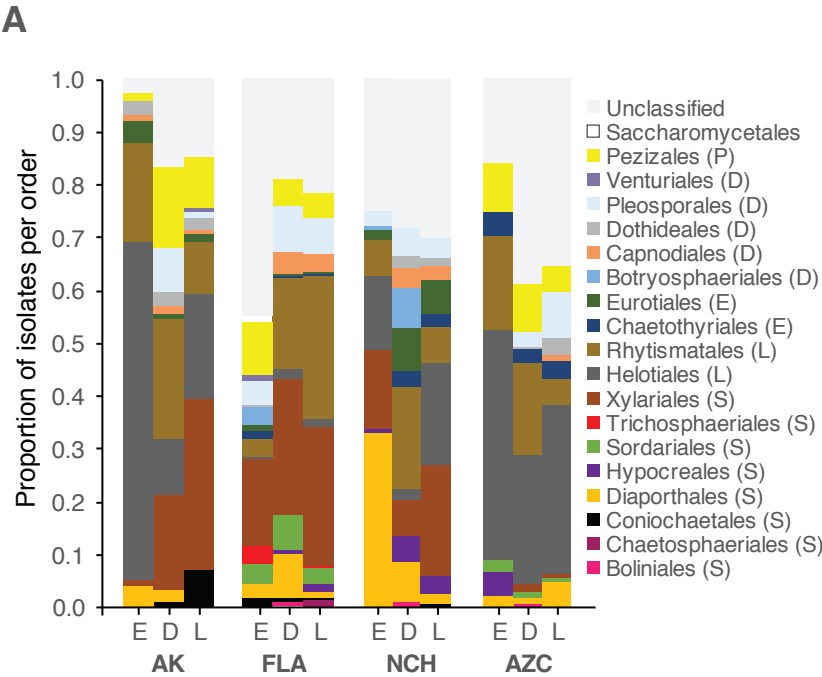

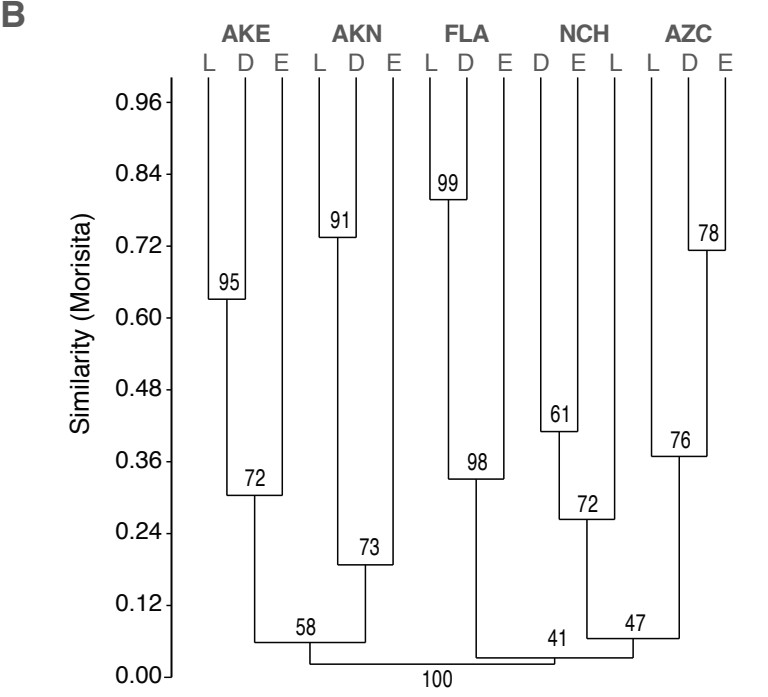

**Figure 2  Order-level taxonomy and hierarchical clustering analysis of fungal communities in living (E), senesced (D), and fallen (L) leaves among sites.** (A) Proportion of isolates representing orders of Pezizomycotina as a function of site/leaf type. Sequences from AKE (Eagle Summit, Alaska) and AKN (Nome, Alaska) were pooled for this analysis (AK). (B) Unweighted pair-group average (UPGMA) dendrogram showing hierarchical clustering of endophytes (Endo), DLF, and LLF from each site (AZC, Chiricahua Mountains, Arizona; NCH, Highlands Biological Station, North Carolina; FLA, Archbold Biological Station, Florida; AK, see above). Bootstrap values are based on 10,000 bootstrap replicates. Host individuals/leaf type with <4 sequences and singleton OTU were excluded from hierarchical clustering analysis.

**Table 4** Results of PERMANOVA analysis of the Morisita-Horn dissimilarities for fungal OTU community structure (DF, degrees of freedom; SS, sum of squares; MS, mean sum of square; F, pseudo-F by permutation).

| | DF | SS | MS | F | $R^2$ | P |
|---|---|---|---|---|---|---|
| Blocks: site (1000 permutations) | | | | | | |
| Host species | 11 | 17.978 | 1.634 | 5.546 | 0.431 | 0.001 |
| Leaf type | 2 | 1.372 | 0.686 | 2.328 | 0.033 | 0.001 |
| Residuals | 78 | 22.398 | 0.295 | | 0.537 | |
| Total | 89 | 41.748 | | | 1.000 | |
| Blocks: site; plots: host species (200 permutations) | | | | | | |
| Leaf type | 2 | 1.446 | 0.723 | 1.560 | 0.035 | 0.005 |
| Residuals | 87 | 40.303 | 0.463 | | 0.965 | |
| Total | 89 | 41.748 | | | 1.000 | |

We focused on OTU in two classes to evaluate the relationship of substrate utilization to leaf type of origin. Dothideomycetes from living leaves grew more rapidly than those isolated originally from non-living leaves on all three carbon sources (Fig. 5C). However, growth of Sordariomycetes on cellulose, lignin, or pectin did not differ for fungi from living vs. non-living tissues (Fig. 5C).

We found no significant correlation between genetic distance and differences in growth on cellulose or lignin, but differences in fungal growth on pectin were correlated with genetic distance between isolates (Table S3).

## Enzyme activity

Extra-hyphal clearing (indicative of enzyme activity) was observed for isolates only on media containing cellulose as the sole carbon source. Only the positive control displayed *in vitro* ligninolytic or pectinolytic activity.

The number of OTU with detectable cellulolytic activity vs. no cellulolytic activity did not differ as a function of leaf type of origin (Table S3). For OTU with detectable clearing, the degree of cellulolytic activity differed significantly among fungi from different classes, with Leotiomycetes from non-living leaves displaying the greatest enzymatic activity (Table 5; Fig. 5A). Dothideomycetes from non-living leaves had greater cellulolytic activity compared to those isolated only from living leaves (Fig. 5B). Sordariomycetes from different leaf types did not differ significantly in cellulolytic activity (Fig. 5B). We found no significant correlation between genetic distance and differences in cellulolytic activity (Table S3).

## DISCUSSION

The occurrence of some endophytic fungi in dead leaves and/or leaf litter and the ability of some endophytes to produce extracellular leaf-degrading enzymes in a manner consistent with saprotrophs (*Carroll & Petrini, 1983*; *Fröhlich & Hyde, 1999*; *Promputtha et al., 2007*; *Korkama-Rajala, Müller & Pennanen, 2008*; *Promputtha et al., 2010*; *U'Ren et al., 2010*; *Chaverri & Gazis, 2011*; *Osono & Hirose, 2011*; *Sun, Guo & Hyde, 2011*; *He et al., 2012*) has

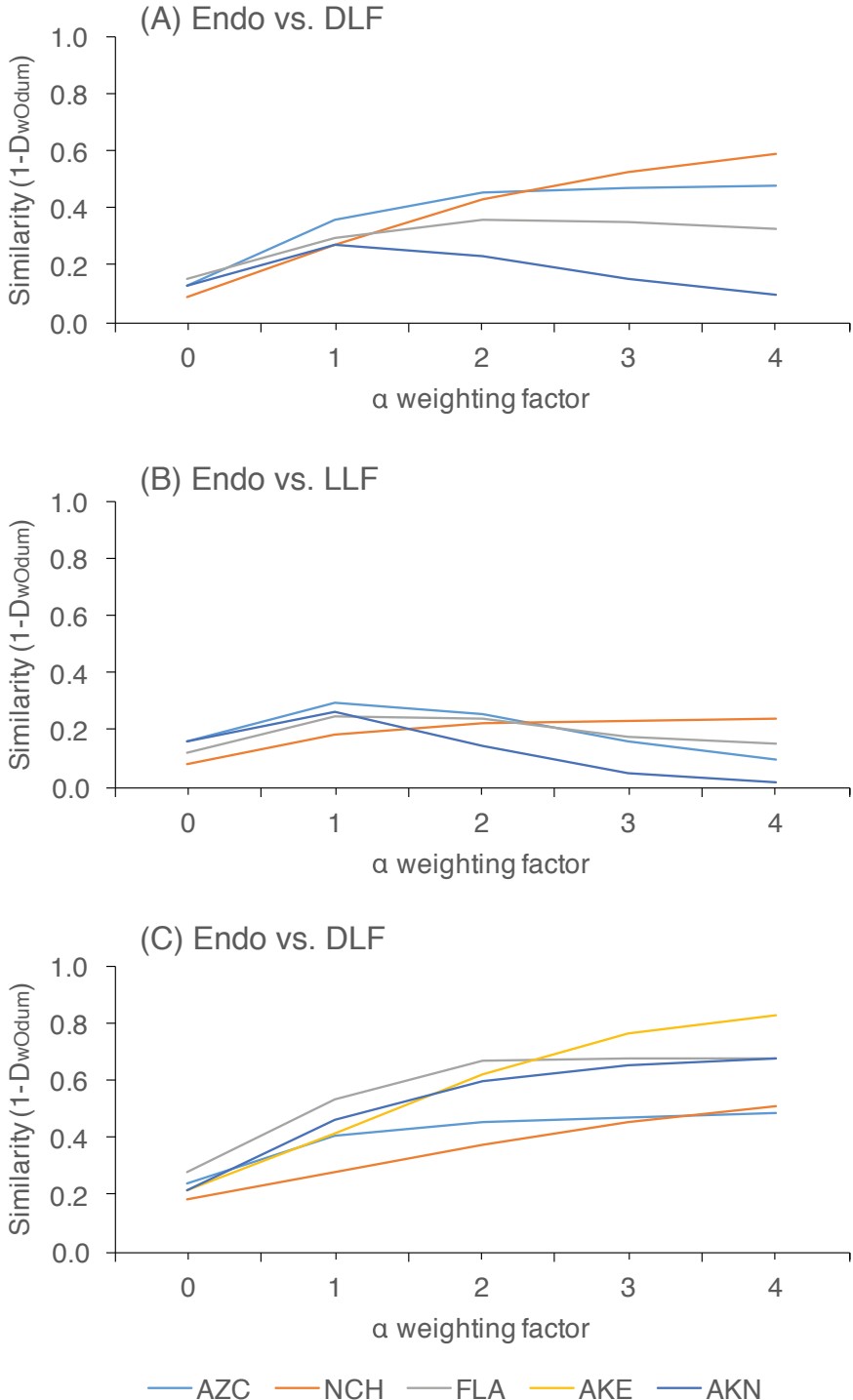

**Figure 3  Pairwise community similarity among different leaf types as a function of site.** Similarity is measured as $1-D_{wOdum}$, as calculated in OTUshuff using 1,000 Monte Carlo iterations. Similarity was assessed at different alpha weighting factors (i.e., when $\alpha = 1$, $D_{wOdum}$ is equal to $D_{Odum}$; when $\alpha < 1$, the influence of abundant taxa will be down-weighted; and when $\alpha > 1$, the influence of low abundance taxa will be down-weighted). Low sample size of endophytes precluded comparisons of endophyte and DLF/LLF at AKE.

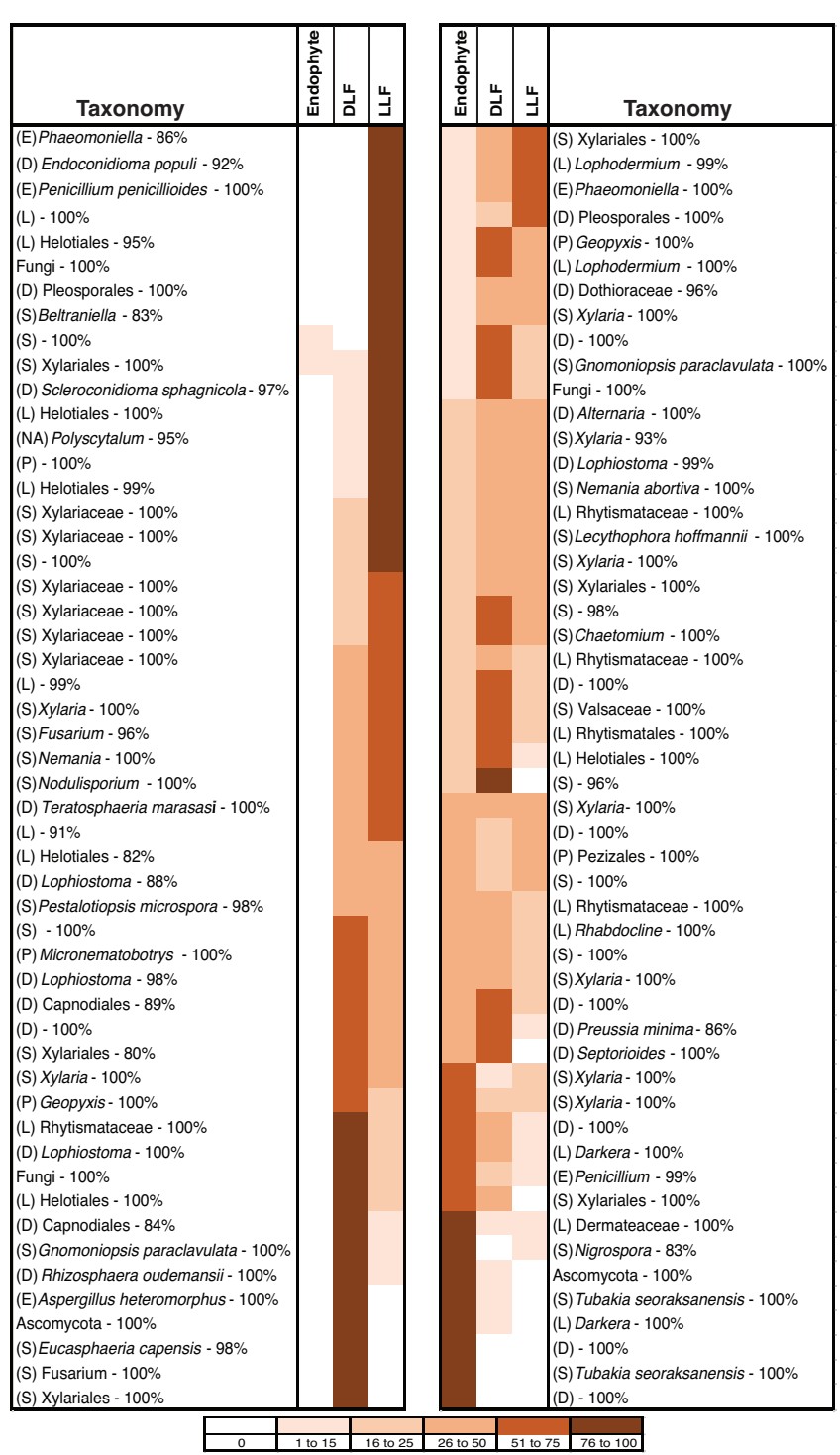

**Figure 4** Heat map showing the distribution of 104 putative species (based on 95% rDNA sequence similarity; OTU with <5 isolates were excluded) of endophytic, dead leaf fungi (DLF), and leaf litter fungi (LLF) from plants in five North American sites. Species abundance on each leaf type is shown as the percentage of the total number of isolates of that species. Taxonomy was estimated for each species using the Ribosomal Database Project naïve Bayesian classifier 

**Figure 4 (…continued)**
with the UNITE ITS database. The lowest level of taxonomy supported by ≥80% confidence is shown
(classes are abbreviated as Sordariomycetes (S), Leotiomycetes (L), Eurotiomycetes (E), Dothideomycetes
(D), Pezizomycetes (P), or not available (NA)). Statistical support for each assignment is given after the
dash. Taxa are ordered according to their abundance as endophytes or LLF.

led to suggestions that many endophytes are incidental symbionts that exist in leaves as
a prelude to completing their life cycles as primary decomposers. Here, we explore the
endophyte–saprotroph continuum, with the goal of understanding the degree to which
endophytes may occur in non-living tissues in diverse plants and terrestrial biomes and
whether ecological patterns reflect fungal carbon substrate utilization and enzyme activity.

## Fungal communities differ in living and non-living leaves

As a whole, culturable fungi inhabiting living leaves were isolated less frequently and were
less diverse than those in non-living leaves. This may reflect the plant actively restricting
fungal colonization within leaves, limited growth by fungi within living leaves for intrinsic
reasons, or culturing biases (see below). Culturable fungal communities in living leaves
also differed in composition from communities in senesced leaves and leaf litter, regardless
of differential weighting of rare and abundant taxa.

These observations are congruent with colonization of senesced and abscised leaves
by additional epiphytic- and/or litter fungi (see *Lindahl et al., 2007*). As leaves senesce,
epiphytic fungi can colonize the interior of leaves rapidly, leading to a subsequent decline
in the relative abundance of endophytic taxa (*Cabral, 1985*; *Stone, 1987*; *Osono, 2002*; but
see *Peršoh et al., 2013*). When we examine common OTU as a function of their abundance
in leaves of each type, 35% have patterns of leaf-association that match that of ambient,
non-endophytic fungi colonizing leaf interiors after leaf death (Fig. 4). Additional fungi
appear to colonize fallen leaves from the underlying litter layer or from other sources; these
represented a relatively small number (8% of OTU), potentially reflecting preferential
growth of Ascomycota (vs. Basidiomycota and other fungal taxa) on the culture medium
used here.

Although culturable ascomycete communities as a whole differed significantly as a
function of leaf type, 29% of non-singleton OTU (Fig. S3) and 52% of common OTU (OTU
represented by >5 sequences) from living leaves also inhabited non-living leaves. These OTU
typically were present in living vs. non-living leaves at different abundances (Fig. 4) This
is consistent with previous studies that noted the transitory nature of many endophytes in
leaf litter (*Stone, 1987*; *Hudson, 1968*; *Osono & Takeda, 2001*), as well as the quick turnover
in litter of all phyllosphere fungi (including both epiphytes and endophytes). For example,
*Voříšková & Baldrian (2013)* found that the majority of phyllosphere OTU were absent
from leaves two to four months after abscission. In contrast, a greater proportion of
Xylariaceae appear to inhabit living leaves, as well as decomposing leaves, wood, and fruit
(*U'Ren et al., 2016*). Factors that mediate these shifts merit further exploration, including
the exhaustion of readily available sugars, the ability of some fungi to decompose structural
polymers, and processes relevant to competitive exclusion or antibiosis (see *Osono, 2006*;
*Yuan et al., 2011*; *Yuan & Chen, 2014*).

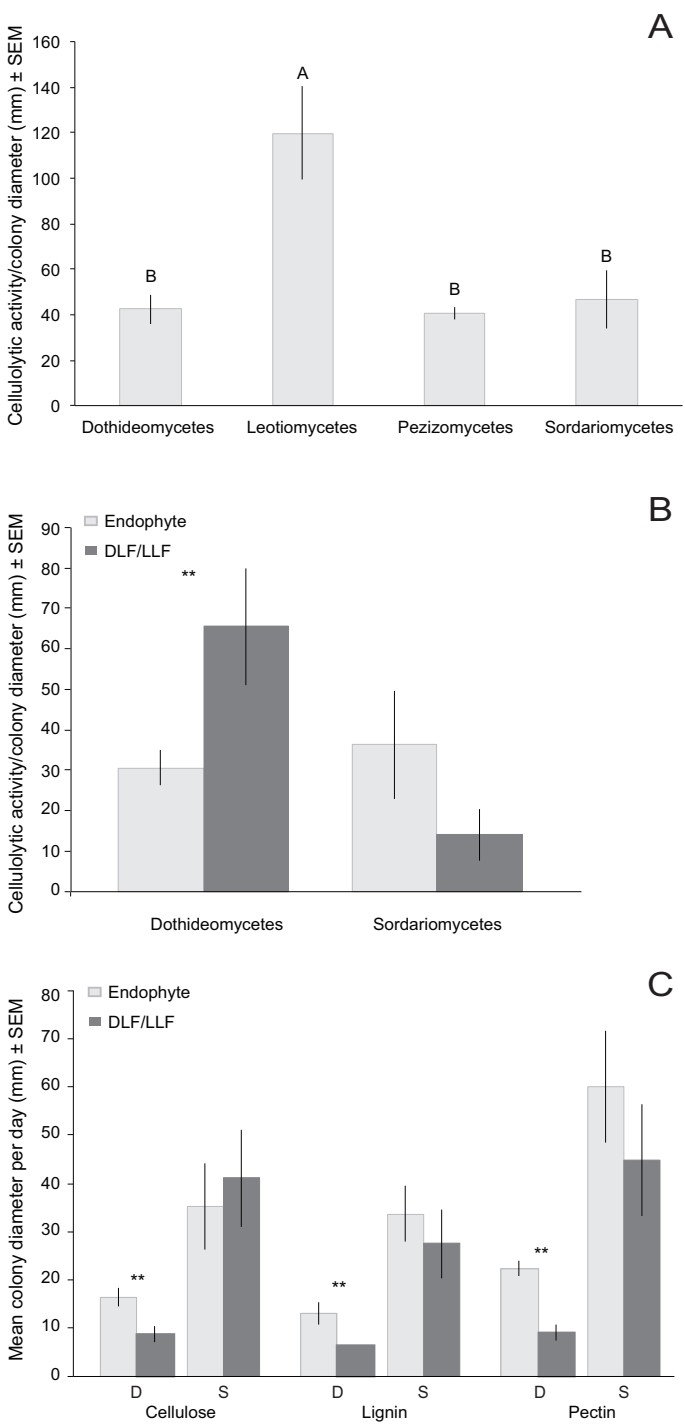

**Figure 5** **Normalized *in vivo* cellulolytic activity (mean activity/colony diameter) and growth (mean colony diameter/day) of 17 representative fungal OTU isolated from living (i.e., endophyte) and non-living leaves (DLF/LLF).** (A) Mean cellulolytic activity as a function of class-level taxonomy and (B) class-level taxonomy and leaf type. (C) Mean growth rate as a function of both class-level taxonomy (S, Sordariomycetes; D, Dothideomycetes) and carbon source 

**Figure 5 (…continued)**
(cellulose, lignin, or pectin). Different letters represent significant differences in enzyme activity as a function of fungal class after *post-hoc* nonparametric multiple comparisons (A). Asterisks (**) indicate significant differences in cellulolytic activity or growth between fungal OTU found only in non-living leaves vs. fungal OTU isolated from only living leaves ($P < 0.05$; Table S3) (B and C). No significant differences were observed between OTU of Sordariomycetes from different leaf types (Table S3) (B and C).

## Differences in fungal communities as a function of leaf type are consistent among sites and host lineages

Although the same major lineages of Ascomycota were present in all sites, fungal communities differed among sites in terms of the relative abundance of those lineages, as well as at finer taxonomic levels. In some cases, these differences may reflect unequal recovery of fungi from different plant lineages: in AKN and AZC, for example, fungi were more commonly isolated from coniferous hosts, and isolation frequency from angiosperms was low compared to other sites. Yet, communities still differed between sites when lineages were sampled equally (e.g., NCH and FLA), consistent with our previous results on endophytic and endolichenic fungi from the same locations (*U'Ren et al., 2012*) and with other studies illustrating the influence of climate and biogeographic factors on endophyte community composition (e.g., *Davis & Shaw, 2008*; *Zimmerman & Vitousek, 2012*; *Langenfeld et al., 2013*).

Within each site we found that fungal community composition differed markedly among host species. These results are consistent with the view that plant defenses may mediate infection and colonization by particular endophytes (see *Schulz & Boyle, 2005*), with persistent signatures of host chemistry or other traits contributing to the structure of fungal assemblages in non-living leaves. Previous work has shown that litter traits (i.e., nutrient quality or other traits relevant to host species) may have stronger effects on fungal community composition than other environmental factors (*Aneja et al., 2006*; *Šnajdr et al., 2011*; *Bray, Kitajima & Mack, 2012*; *Urbanová, šnajdr & Baldrian, 2015*). More generally, the consistent differences observed between fungal communities in living vs. non-living leaves in different hosts and sites suggest that factors influencing the ecology and evolution of plant-fungal associations may be relatively consistent at large scales.

## Evaluating the strength of our conclusions

Differences in fungal communities observed here could result from artifacts of insufficient sampling, unintentional comparisons among leaves of different ages (e.g., leaf litter representing previous years of growth vs. living leaves representing the current year), and/or biasing our work toward fungi that are readily cultured on standard nutrient media or specific isolation techniques (see *Unterseher & Schnittler, 2009* for comparison of fragment plating vs. dilution-to-extinction culturing). The last issue might be especially problematic for fungi that do not normally occur in symbiosis and/or have the capacity to grow on non-living substrates vs. those with a symbiotrophic lifestyle. However, several lines of evidence suggest our conclusions are robust. First, species accumulation curves and bootstrap estimates of species richness for all sites and site/leaf-type combinations indicate our sampling was at or near statistical completion for culturable fungi. Second,

**Table 5** *In vitro* cellulolytic activity and growth on cellulose, lignin, and pectin for 17 representative isolates of OTU found only in living (i.e., endophytes) or non-living leaves (i.e., DLF/LLF).

| Isolate Name | 95% ITS-partial LSU OTU | Substrate type | Isolates in OTU[a] | Sites[a] | Plant host species (families)[a] | Taxonomy[b] | Mean ± SD growth rate on cellulose medium (mm/day) | Mean ± SD growth rate (mm/day) on lignin medium | Mean ± SD growth rate (mm/day) on pectin medium | Mean ± SD cellulolytic activity/colony diameter (mm) |
|---|---|---|---|---|---|---|---|---|---|---|
| NC0075 | 403 | Living | 26 | 1 | 2 (2) | Dothideomycetes (*Phyllosticta**) | 10.7 ± 1.1 | 7.2 ± 1.0 | 25.0 ± 3.1 | 22.0 ± 6.6 |
| NC0101 | 404 | Living | 6 | 1 | 2 (2) | Dothideomycetes (*Phyllosticta**) | 11.4 ± 2.0 | 7.8 ± 0.4 | 22.1 ± 3.8 | 40.0 ± 16.8 |
| FL0319 | 261 | Living | 7 | 1 | 1 (1) | Dothideomycetes (*Ochrocladosporium**) | 17.1 ± 0.3 | 10.4 ± 0.3 | 14.5 ± 0.4 | 45.3 ± 1.5 |
| FL1985 | 366 | Non-living | 13 | 1 | 2 (2) | Dothideomycetes (*Penidiella**) | 7.3 ± 0.3 | 6.2 ± 0.5 | 9.6 ± 0.6 | 49.0 ± 6.0 |
| FL0303 | 259 | Living | 12 | 1 | 1 (1) | Dothideomycetes (*Mycosphaerella**) | 26.7 ± 0.4 | 26.9 ± 1.6 | 28.3 ± 0.8 | 15.3 ± 4.0 |
| FL1704 | 353 | Non-living | 9 | 1 | 2 (1) | Dothideomycetes (*Teratosphaeria marasasii*) | 10.2 ± 5.8 | 6.6 ± 0.4 | 8.5 ± 0.1 | 82.0 ± 42.5 |
| AK1907 | 118 | Non-living | 16 | 3 | 4 (1) | Leotiomycetes (*Hyalodendriella**) | 11.3 ± 0.2 | 11.7 ± 0.9 | 12.6 ± 0.3 | 46.0 ± 2.0 |
| FL2076 | 360 | Non-living | 9 | 1 | 2 (2) | Leotiomycetes (*Rhytisma**) | 6.2 ± 1.4 | 6.8 ± 0.5 | 6.2 ± 1.6 | 145.7 ± 38.8 |
| FL2145 | 355 | Non-living | 7 | 1 | 1 (1) | Leotiomycetes (*Rhytisma**) | 7.0 ± 0.8 | 4.1 ± 1.3 | 23.1 ± 1.2 | 167.7 ± 28.9 |
| AZ0245 | 163 | Non-living | 7 | 1 | 2 (2) | Pezizomycetes (*Strobiloscypha**) | 13.2 ± 0.5 | 5.9 ± 0.4 | 31.3 ± 3.3 | 40.3 ± 4.6 |
| FL0231 | 253 | Living | 7 | 2 | 5 (2) | Sordariomycetes (*Lecythophora**) | 20.5 ± 0.6 | 25.9 ± 1.1 | 22.8 ± 1.0 | 0.0 ± 0.0 |
| NC0063 | 406 | Living | 7 | 1 | 4 (3) | Sordariomycetes (*Eutypa**) | 25.2 ± 1.3 | 21.7 ± 0.6 | 21.0 ± 0.3 | 28.3 ± 4.5 |
| NC0012 | 399 | Living | 12 | 1 | 1 (1) | Sordariomycetes (*Tubakia seoraksanensis*) | 9.0 ± 0.2 | 20.3 ± 1.2 | 105.0 ± 4.4 | 116.7 ± 4.9 |
| FL2151 | 352 | Non-living | 31 | 1 | 4 (3) | Sordariomycetes (*Magnaporthe**) | 24.4 ± 0.3 | 20.2 ± 0.2 | 21.2 ± 0.5 | 0.0 ± 0.0 |
| NC1320 | 378 | Non-living | 6 | 2 | 2 (2) | Sordariomycetes (*Fusarium**) | 81.0 ± 4.4 | 55.2 ± 2.9 | 91.0 ± 1.7 | 3.0 ± 0.0 |
| FL1642 | 170 | Non-living | 13 | 3 | 3 (1) | Sordariomycetes (*Polyscytalum*) | 17.8 ± 1.3 | 6.9 ± 1.1 | 21.9 ± 2.4 | 39.0 ± 5.6 |
| NC0068 | 263 | Living | 13 | 2 | 2 (2) | Sordariomycetes (*Collectotrichum**) | 86.3 ± 0.6 | 66.7 ± 0.8 | 91.3 ± 0.6 | 0.0 ± 0.0 |

**Notes.**

[a]Totals include fungi isolated from lichens and plants as part of a larger study on endophytic and endolichenic fungi (*U'Ren et al., 2012*)

[b]Taxonomic information is based on queries of UNITE database using the RDP Bayesian classifier (see methods). However, when matches to the UNITE database lacked genus and species information (e.g., the species hypothesis for FL1985 is Capnodiales sp.), taxonomic information is estimated based on a BLASTn query of the NCBI nr database (indicated with an *).
although leaves may have emerged at different times and thus been subject to different pools of inoculum (see *Arnold & Herre, 2003*), patterns are consistent across hosts from different microsites, geographic locations, and host species, including both deciduous and evergreen angiosperms as well as conifers. Third, although missing species found using culture-independent methods have the potential to alter our estimates of diversity and the degree of overlap among leaf-types (*Arnold et al., 2007*; *Gallery, Dalling & Arnold, 2007*; *Jumpponen & Jones, 2009*; *U'Ren et al., 2014*; *Balínt et al., 2015*), the consistent patterns we observed across hosts and sites suggest that our conclusions are valid for prevalent members of these fungal communities. In addition, our results are congruent with similar studies that used culture-free methods (*Voříšková & Baldrian, 2013*).

The abundance and identity of endophytes in leaf litter typically differs as a function of leaf age and season (*Hirose & Osono, 2006*; *Osono, 2006*; *Peršoh et al., 2013*). Had we sampled freshly fallen leaves, communities may have been more similar among leaf-types; conversely, greater differences may have been observed had we examined more thoroughly decomposed leaves (see *Voříšková & Baldrian, 2013*). Accordingly, our estimates of overlap among endophytic and saprotrophic communities are restricted to culturable members from leaf litter in a moderate state of decay sampled during the summer. However, the relatively consistent patterns observed from tundra to subtropical communities suggest strong signal in the data.

Importantly, the relative abundance of isolates in living or dead tissues is not necessarily indicative of an isolate's ability to decompose plant tissues. This prompted our evaluation of substrate use and enzyme activity (below). Future studies that assess endophyte and litter communities over multiple time points (including multiple time points for living leaves and during the first weeks of decomposition) in combination with functional assays and metatranscriptomic methods, will provide much-needed insight into the persistence and functional roles of endophytes as decomposers.

## Functional differences

The limited persistence of endophytes in leaf litter over time led us to conduct a preliminary investigation of *in vitro* patterns of substrate use between OTU found only in living tissues vs. only in non-living tissues in our surveys (including the larger sampling from *U'Ren et al., 2012*). Across four classes of Pezizomycotina, cellulolytic activity differed primarily as a function of class-level taxonomy. However, cellulolytic activity was not correlated with finer-scale genetic relationships within Sordariomycetes and Dothideomycetes. Although some isolates demonstrated high levels of cellulolytic activity (e.g., Leotiomycetes representing the Rhytismataceae and Helotiaceae), our study differs from previous ones in that none of our isolates had detectable ligninolytic or pectinolytic activity (see *Oono et al., 2014*; *Fouda et al., 2015*). We detected differences in cellulolytic activity and growth rate on pectin-, lignin-, and cellulose media between Dothideomycetes fungi from living vs. non-living leaves. However, such differences were not observed in the Sordariomycetes. Overall, more precise measurements investigating a broader array of enzymes (e.g., $\beta$-mannanase, xylanase, laccase, etc.), in conjunction with genomic and transcriptomic studies of multiple

isolates per OTU, are necessary to identify functional differences among fungi associated with leaves at different life stages.

## Conclusions

Approximately half of the common OTU found here occurred in both living and non-living leaves. However, our sampling across diverse biomes and host lineages detected consistent differences in living vs. non-living leaves in terms of diversity and fungal community composition. Fungi that were common in decomposing leaves collected from leaf litter were not typically common in living leaves of the same plants. Thus some fungi with endophytic life stages clearly persist for periods of time in leaves after senescence and incorporation into leaf litter, with shifts after leaf death and in the early stages of decomposition likely reflecting incursion by additional fungi from the leaf exterior. Overall, our data—while preliminary in some regards—suggest that focal isolates found in dead and fallen leaves did not differ consistently from those in living leaves in terms of their capacity to use pectin, lignin, and cellulose as their sole carbon sources, nor in the degree of cellulolytic activity displayed *in vitro*. We suggest that culturable fungi with endophytic life stages may not persist in decomposing leaves once cellulose and hemicellulose are depleted. Future analyses should optimize methods to culture both Basidiomycota and Ascomycota endophytes, and should explore more dimensions of functional traits and persistence to further define the endophytism-to-saprotrophy continuum.

## ACKNOWLEDGEMENTS

We thank E Gaya, K Molnár, T Abbey, K Arendt, F Santos, M Gunatilaka, M Hoffman, M. del Olmo R, M Orbach, DL Taylor, and especially F Lutzoni and J Miadlikowska for field and laboratory assistance, DR Maddison for sharing pre-release versions of Mesquite and ChromaSeq, P Degnan, T Wheeler, J Stajich, S Huse, SJ Miller, Y Sun, and N Zimmerman for computational assistance, and RJ Steidl and R Kaczorowski for assistance with statistical analyses. We gratefully acknowledge the NSF Fungal Environmental Sampling and Informatics Network (FESIN; DEB0639048 to T Bruns, K Hughes, and AEA) for fostering discussion that informed this work, and N Zimmerman and five anonymous reviewers for helpful comments on the manuscript.

### Funding

This work was funded by the National Science Foundation (DEB0640996 and DEB072825 to AEA and an NSF-supported IGERT Fellowship in Genomics to JMU). JMU also was supported by the Mycological Society of America's Clark T. Rogerson student research award and Graduate Fellowship. The funders had no role in study design, data collection and analysis, decision to publish, or preparation of the manuscript.

## Grant Disclosures

The following grant information was disclosed by the authors:

National Science Foundation: DEB0640996, DEB072825.

NSF-supported IGERT Fellowship in Genomics.

Mycological Society of America Clark T. Rogerson student research award.

Mycological Society of America Graduate Fellowship.

## Competing Interests

The authors declare there are no competing interests.

## Author Contributions

- Jana M. U'Ren conceived and designed the experiments, performed the experiments, analyzed the data, wrote the paper, prepared figures and/or tables, reviewed drafts of the paper.
- A. Elizabeth Arnold conceived and designed the experiments, contributed reagents/materials/analysis tools, wrote the paper, reviewed drafts of the paper.

## Field Study Permissions

The following information was supplied relating to field study approvals (i.e., approving body and any reference numbers):

All fungal samples relevant to this work were handled in accordance with standard operating procedures for USDA permit P526P-1400151. No specific permits were required for collecting.

## DNA Deposition

The following information was supplied regarding the deposition of DNA sequences:

GenBank accession numbers are provided for all sequences in Table S2.

## Data Availability

The raw data have been supplied as a Supplemental Dataset.

## Supplemental Information

Supplemental information for this article can be found online at http://dx.doi.org/10.7717/peerj.2768#supplemental-information.

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
