# Peer review of "Diversity, taxonomic composition, and functional aspects of fungal communities in living, senesced, and fallen leaves at five sites across North America"

_PeerJ, doi:10.7717/peerj.2768_

## Round 0.1 · original submission · Major Revisions

I enclose reports from three referees who are experts in microbial ecology of plant endophytes.  They have commented favorably on the relevance and novelty of your paper (see below) -- and I fully agree with them. You should be commended for the ambitious geographical range and the broad selection of hosts - neither of which have been satisfactorily addressed yet.

The referees however have a number of queries, concerns and suggestions indicating that the current work suffers from several methodological and editorial weaknesses: e.g.,

- The choice of the diversity indices should be better justified;
- A focus on the persistence of foliar endophytes in senescent leaves and leaf litter would ease the reading of your study;
- Mixing review style with meta-analysis and new data is detrimental to flow of the paper;
- Already published sequence information (supplementary table 1) cannot be published as it stands. I agree with referee #1: 'The entire manuscript needs considerable shortening to avoid identical information in two different publications.'

I am also concern by referee #1's comment: 'It further lacks convincing evidence that the data really support the results'. BTW, Please note that referee #1 has attached a document with his/her edits and questions.

I really think that you should comply with referees' requests and provide the requested clarifications to make this paper stronger and in line with the high standard of PeerJ.

Reviewer 1 ·

Basic reporting

In the present manuscript entitled "Diversity, distributions, and host associations of fungal communities in living, senescent, and fallen leaves at five sites across North America", the authors U'Ren and Arnold provide a profound, deeply referenced and comprehensive manuscript". While acknowledging the immense effort put into the present manuscript I have to comment on some major shortcomings. Starting with the introduction a lack of most recent references is obvious. Only 5 studies younger than 2014 are cited thus neglecting most recent substantial progress in this field. Substantial effort is needed to replace some to many of the most archaic studies with recent state of the art papers (e.g. more of high-throughput studies) of plant-associated fungi. Given the comparatively long introduction, several subheadings might be helpful.

In summary the entire manuscript suffers from missing clarity, because review style is mixed with meta-analysis and new data. It further lacks convincing evidence that the data really support the results (see below), as some prerequisites mentioned by the authors themselves (l. 553 ff), were not guaranteed: a) species accumulation curves are not shown for hosts and sites and b) bootstrap estimator alone is not reliable.

My overall rating therefore is major revision. The attached word document contains many more comments and questions.

Experimental design

From the Materials and Methods and Results it becomes obvious that parts of the data were already analysed and published, but to what extent remains widely unclear. Supplementary table 2 for instance should not be printed in its present form, since it contains already published sequence information, supplementary table 1 is identical to that shown in U'Ren et al. 2012 and should removed, too. Much more effort is necessary to clearly identify the proportion of data and analyses new to science. The entire manuscript needs considerable shortening to avoid identical information in two different publications.

Validity of the findings

With respect to diversity and community composition the applied statistics seem widely appropriate and comprehensive. However I have to question the quality of underlying data, especially in terms of completeness. On the one hand the total number of 2618 isolates can be considered intensive sampling. However, if downscaled to 18 host plants it rarely meet the important criteria of exhaustiveness, unless proven by additional accumulation curves and estimators. The use of Bootstrap estimator as single method to determine sampling completeness is not sufficient, at least a member of the Chao family is needed to support the presented results and discussions.

Accumulation curves should be calculated not only for the three substratum types, but also for the five sites and the 18 host plants. Only if they show comparable and exhaustive sampling intensity (e.g. roughly 70% of expected total richness), they can reliably compared.

Additionally, I do not see convincing reasons for excluding rare OTUs (except of difficult statistical handling). Every sequence comes from a real fungal culture and the chromatograms were obviously checked in order to rule out sequencing errors. Given the comparatively low per host sampling intensity (mostly below 100 isolates per category) the meaningfulness of several data and statistics should be critically re-analysed (e.g. in column seven of table 1).

ITS-based taxon annotation at the class level should be considered insufficient, given the substantial progress in ITS barcoding and availability of curated data bases. Efforts of the UNITE consortium are widely neglected although use of their reference data bases provides semi-automated and reliable annotations often to genus and species level for a broad spectrum of plant-associated fungi.

Additional comments

no comments

Annotated reviews are not available for download in order to protect the identity of reviewers who chose to remain anonymous.

Reviewer 2 ·

Basic reporting

No comments

Experimental design

No comments

Validity of the findings

No comments

Additional comments

This manuscript describes an effort to address a question that has remained open despite the focus that endophytes have received: do the endophytic fungi remain in the tissues and convert to a life style more dominated by saprotrophy. The current effort expands on previous studies by incorporating leaves before and after abscission and includes a number of host plants as well as five geographic location. In sum the manuscript describes an ambitious effort to isolate fungi from pre and post-abscission tissues over a large scale and from number of host plants. Importantly, the study also includes an assessment of lignocellulysic enzyme activities and determines that the fungi from living and fallen leaves differ in their enzyme activities.

The manuscript is clearly written and largely uses established approaches and protocols used by the authors earlier. I have only few comments that primarily request few pieces of additional information and pick on some grammatical/typographical issues.

Line 159: leaf types were selected to represent the same year of growth. How is this possible for fallen needles of Pinaceae, which retain their leaves for up to 5 years?

Line 266: I would like more details on the nitrate salts, trace elements and vitamin solution.

Line 268: Company Info.

Line 292: less

Line 293: distance matrices

Lines 327, 329: use plural for OTUs. I believe this is rather commonly used.

Line 355: does cluster refer to ordination? If so, please use the latter as clustering is incorrect.

Line 486: Microbotryomycetes

Reviewer 3 ·

Basic reporting

I enjoyed reading the article by U'Ren & Arnold. It is clearly written, the literature is complete and very relevant. The study presents novel and exciting findings on foliar fungal communities.

Minor comments on the introduction:
Line 111: "Several key questions remain unanswered". I think that the authors should list these questions instead of discussing sampling methods (lines 111-120)
Line 121: The study is presented as a "part of a larger study" that has already led to 4 publications. I am not sure that this sentence is necessary. The introduction is already quite long.
Lines 126-136: I suggest that the authors rephrase the objectives of the study. It is not clear if the study was designed to test specific ecological hypotheses, or if it is rather a large-scale survey of fungal diversity.
Line 130: What is "leaf type"?
Lines 131-132: Climate and host lineage are both expected to have an effect on fungal community structure. A few references on this topic are missing from the introduction. Why do the authors expect that shifts in community structure from living to non-living leaves do not depend on climate and host species?
Lines 132-133: "differ in their carbon utilization patterns". Would it be possible to formulate a more specific hypothesis?

Experimental design

The experimental design is well presented but the statistical section could be improved. Moreover, the number of statistical tests and figures could be reduced. The main goal of the study is to investigate the persistence of foliar endophytes in senescent leaves and leaf litter. I suggest that the authors remove the analyses that are not directly linked to this main objective.

Comments on the materials and methods section:
Lines 246-247: The choice of the diversity indices should be justified. Bray-Curtis index (the quantitative version of the Jaccard index) could have been used as well.
Lines 197-296: Statistical analyses would be easier to assess and understand if they were presented in a single section. Moreover, it seems that the calculation of diversity indices and the statistical tests were performed using various softwares (Estimate S, JMP, mothur, BiodivR, PAST, ade4 in R). I think that the authors should rewrite the statistical section and perform again the analyses using R.

Comments on the results section:
Lines 302-303: What is the average number of sequences (isolates) per OTU? Does the number of sequences per OTU reflect OTU abundance? The data should be transformed in presence-absence if the sampling effort was too low to consider the number of sequences per OTU as a proxy for OTU abundance.
Lines 310: "Leaf type" and "host type" should be better defined.
Lines 314-319: I don't think that these results give an insight into the abundance and diversity of fungi in living versus non-living leaves (lines 422-423). They only show the abundance and diversity of culturable fungi in both leaf types.
Line 334 and Figure 3: The high congruence in endophytic communities is not revealed by Figure 5, why?
Line 310: The effects of the main factors - leaf type, host type and site - on fungal community structure are tested separately. It would be more correct to do a single analysis including the three factors and their interactions.
Line 373-375 and elsewhere: There are too many results in the article. I think that the article could be improved by focusing on the effect of leaf type on community composition and substrate utilization. Other effects (host species, site) could be presented elsewhere.

Validity of the findings

The main result of the study is that foliar endophytes hardly persist in dead lives, and play a limited role in litter decomposition. I think that the authors should shorten the discussion, by focusing on this main finding. The limitations of culture-based methods should also be better discussed.

Comments on the discussion
Lines 422-423: Again, I am not sure that the results give an insight into the abundance and diversity of fungi in living versus non-living leaves (lines 422-423). They only reveal the abundance and diversity of culturable fungi in both leaf types. Saprotrophs could be easier to isolate than endophytes.
Lines 458-459: This result should not be emphasized too much. I suggest that the authors discuss the limitations of culture-based methods.
Lines 518-519: Limitations of culture-based methods are finally discussed. But it is too late in the manuscript.

---

## Round 0.2 · Minor Revisions

The two referees have commented favorably on the relevance of your paper (see below), but suggested editorial changes to produce a stronger article. Despite the concerns about the science and over presentation, I believe your manuscript has strong potential to be a valuable addition to the field following careful revision. I am therefore recommending it be returned for minor revisions.

Reviewer 4 ·

Basic reporting

The term "senescent leaves" is misleading, because from the text it appears that instead the meaning is "dead leaves", not old but still living (i.e. senescent) leaves. The title and abstract should be modified accordingly.

Experimental design

No Comments

Validity of the findings

Xylariaceae have often been reported as endophytes in leaves, but very rarely as decomposers of dead leaves. Is the reason that a comparison between Xylariaceae in living and dead leaves is not made here that the authors have published about endophytic Xylariaceae in another paper?

The authors come to the conclusion that “in the long term” endophytes are replaced by secondary saprobic invaders, but could the implicit “short term” (how many weeks?) be discussed?

Since the authors tend to exclude the pioneer saprobic role of endophytes (“in the long term”), which other possible roles may be played by the endophytes?

Additional comments

I see that the manuscript has already been very carefully reviewed and revised so that I do not need to add new detail questions, but mainly general remarks. The scientific question and the approach are highly significance, because most studies of endophytic fungi find mainly ubiquitous saprobic and latent pathogenic fungi. The role of these endophytes might be pioneer decomposers after the dead of the plant (or the leaf as in the present study) and competition avoidance against secondary saprobic invaders, but this hypothesis has not really been directly addressed experimentally. This study, however, directly compares the diversity of cultivable fungi from living leaves, dead leaves attached to the tree, and from fallen dead leaves on systematic base. The molecular identification is based on the usual routine ITS barcode, whose interpretation is particularly modified with a strict (100%) and vague (95%) similarity threshold instead of the routine 97% threshold. The data are connected with another new dataset about the degradation abilities of the isolates.

However, the study deals in terms of input (fungi in living leaves) and output (fungi of dead leaves), but does not address the fungal species. I do not share the view of the authors and other reviewers that ecological phenomena can be adequately expressed in terms of data analysis of input and output alone without identification of the species. It is not even attempted to provide species names in the lists/tables. In contrast to bacteria, fungal classification is not based on physiological properties so that mentioning taxa above the genus level is of very limited ecological relevance. Lack of species names will drastically reduce the potential of citations, because re-usable and thus citable data about species are missing.

ITS and LSU rDNA sequences are not sufficient to distinguish between species of the most common genera occurring on plants, such as Aspergillus, Colletotrichum, and Fusarium. Each of these genera requires additional specific protein gene markers for species distinction. Different species of these genera are likely to be lumped together to one species with the primers selected in this study so that species numbers are difficult to evaluate. But this is a problem common to almost all papers about endophytes. Although a high proportion of mycelia remain sterile in culture, some identifications could be further confirmed or modified by morphological investigation.

Since the authors spent a lot of efforts for the degradation abilities of the strains, it is very pity that we do not know which species are involved. The species specific growth in the leaf during different stages of leaf development is addressed in the Introduction, but not by the experimental setting.

The above comments may considered in future studies, because it would be too much trouble to generate meaningful species lists. Below I only pick up some topics which should be addressed in the Discussion:

The term "senescent leaves" is misleading, because from the text it appears that instead the meaning is "dead leaves", not old but still living (i.e. senescent) leaves. The title and abstract should be modified accordingly.

Exceptions from fungi identified above the species level and still present some ecological significance are the Xylariaceae which have often been reported as endophytes in leaves, but very rarely as decomposers of dead leaves. Is the reason that a comparison between Xylariaceae in living and dead leaves is not made here that the authors have published about endophytic Xylariaceae in another paper?

The authors come to the conclusion that “in the long term” endophytes are replaced by secondary saprobic invaders, but could the implicit “short term” (how many weeks?) be discussed? Since the authors tend to exclude the pioneer saprobic role of endophytes (“in the long term”), which other possible roles may be played by the endophytes?

In my opinion, focusing on a single plant species with fewer localities, but aiming at real species discrimination in the fungi would allow better conclusions than the approach with masses of superficial data and a subsequent highly sophisticated analysis. This does not mean that this is not a good study compared to other similar ones. The problem is that there are many studies about endophytic fungi, which all use the same approaches so that substantially new discoveries are rarely reported. Compared to the average endophyte study, the paper does not only excel by the masses of data, but also qualitatively by the comparison between living and dead leaves and analyzing degradation capabilities.
If the authors address my points of concern in their discussion as they already did with other fundamental limitations (such as the cultivability), the data will present a convincing approximation under the limited conditions of the experimental setting. It will be up to other researchers to use a more organismic approach in order to challenge the data presented here. Therefore, I recommend minor revision of the Discussion Section and subsequent quick publication in order to open the data and conclusions to discussion among the scientific community.

Reviewer 5 ·

Basic reporting

The manuscript by U’Ren and Arnold is a comprehensive analysis detailing the diversity of culturable fungi detected in living, senescent and fallen leaves of plants. They have extensively isolated fungi from surface-sterilized leaf sections and characterize the diversity and the composition of the fungal isolates as well as their ‘decay’ potential using in vitro activity tests.

Experimental design

I think the sampling is impressive and the derived culture collection quite comprehensive. The statistical analysis looks adequate. However, I have listed some major concerns below that must be addressed or at least clarified.

I am not convinced by the idea to cluster ITS sequences into OTUs for culture-dependent studies. I do not see the point since the high quality ITS sequences are derived from pure fungal cultures. Therefore authors loose resolution power (306 OTUs vs. 555 unique genotypes, line 305-306). Authors could have done exactly the same analysis without performing OTU clustering. I really don’t understand the strategy and ask the authors to comment on that.

It would have been interesting to use the same leaf samples and perform culture-independent community profiling to calculate the proportion of OTUs (valid in this case) that have a representative isolate as pure culture (i.e. fungal recovery rate from leaves). This strategy would more convincingly show to which extent their comprehensive fungal cultures reflect the natural fungal diversity (abundant taxa) detected in leaves.

Authors have cultured some Basidiomycete taxa but these were excluded for unclear reasons. Authors must comment on that.

I think that a phylogenetic tree showing the taxonomic diversity of all fungal isolates as well as the corresponding information regarding the sample-types and the location is essential here. Some user-friendly tools like iTOL (http://itol.embl.de/) can greatly improve data visualization.

Only 17 representative fungi have been use for in vitro cellulolytic analysis. This number is not enough to draw robust conclusions regarding the ‘decay potential’ of leaf-associated fungi (i.e endophytes vs. fungi in non-living leaves)

Validity of the findings

No comments (see Experimental Design section)

Additional comments

Other minor issues

The fact that the diversity of cultured endophytes is less abundant than in fallen leaves is interesting and suggests that 1) the plant immune system may restrict fungal diversity in living leaves and/or 2) fungal endophytes are more difficult to retrieve as pure culture than fungi from fallen leaves due to host dependency. Author must include these alternative hypotheses in the discussion section.

In the title, I would use culturable fungi instead of fungal communities.

Figure 2 and 3: define AZC, NCH, FLA, AKE and AKN

I have not checked the typos

---

## Round 0.3 · accepted · Accept

It's a very nice paper and I should thank you and Betsy for responding with such high professionalism to the critiques. Your attention to the extra experiments and details demanded were exemplary and I hope that you and your collaborators think that the ultimate product is better.

Again, congratulations!